# A Mosaic Method for Side-Scan Sonar Strip Images Based on Curvelet Transform and Resolution Constraints

**DOI:** 10.3390/s21186044

**Published:** 2021-09-09

**Authors:** Ning Zhang, Shaohua Jin, Gang Bian, Yang Cui, Liang Chi

**Affiliations:** Department of Military Oceanography and Hydrography and Cartography, Dalian Naval Academy, Dalian 116018, China; nick_zn180@163.com (N.Z.); trighosts@163.com (G.B.); 13998435151@163.com (Y.C.); liangchi202107@163.com (L.C.)

**Keywords:** strip images mosaic, image resolution, curvelet transform, image fusion

## Abstract

Due to the complex marine environment, side-scan sonar signals are unstable, resulting in random non-rigid distortion in side-scan sonar strip images. To reduce the influence of resolution difference of common areas on strip image mosaicking, we proposed a mosaic method for side-scan sonar strip images based on curvelet transform and resolution constraints. First, image registration was carried out to eliminate dislocation and distortion of the strip images. Then, the resolution vector of the common area in two strip images were calculated, and a resolution model was created. Curvelet transform was then performed for the images, the resolution fusion rules were used for Coarse layer coefficients, and the maximum coefficient integration was applied to the Detail layer and Fine layer to calculate the fusion coefficients. Last, inverse Curvelet transform was carried out on the fusion coefficients to obtain images in the fusion area. The fusion images in multiple areas were then combined in the registered images to obtain the final image. The experiment results showed that the proposed method had better mosaicking performance than some conventional fusion algorithms.

## 1. Introduction

As the depth of global ocean exploration continues to increase, understanding the seafloor surface and near-surface is of great significance in the “digital ocean” and “transparent ocean” era. Currently, side-scan sonar is an important means to explore seafloor geomorphology [1], and side-scan sonar images provide important data for seafloor object identification, classification of seafloor sediments, and exploration of marine resources [2,3]. In order to obtain side-scan sonar images in the entire testing zone, the most important task is to mosaic the strip images, in addition to seafloor tracking, slant-range correction, gain correction, and geocoding [4]. Side-scan sonar is generally operated using towing cables, which leads to inaccurate location information. If the coordinate information is directly used to mosaic the images, there will be distortion in the images [5,6,7,8]. Currently, a large number of studies have been carried out to achieve a mosaic of object-level strip images that produce images with complete information and of high quality.

By dividing sonar strip images into paired objects and shadows, Daniel et al. [9] realized rigid registration of side-scan sonar images using a decision tree. Through region segmentation, Thisen et al. [10] extracted shadow areas from side-scan sonar images and calculated the displacement between two images using the cross-correlative method, thereby achieving rigid registration. Vandrish et al. [11] showed that the scale invariant feature transform (SIFT) algorithm can be used for registration of sonar images, although the accuracy was not ideal. Using correlation coefficients and mutual information as similarity parameters, Chailloux et al. [12] extracted a series of significantly correlated areas on adjacent strip images and calculated the global rigid transformation parameters and local elastic transformation parameters, thereby eventually realizing mosaic of adjacent strip images. Wang et al. [13] improved the pre-processing method of side-scan sonar images to extract feature points more accurately and effectively after preprocessing; they also proposed a sped up robust feature (SURF)-based elastic mosaic algorithm to achieve feature-level conformal mosaic of the images. Moreover, Cao et al. [14] used wavelet transform in a strip image mosaic, yet it required the 3D posture information of the side-scan sonar. Zhao et al. [15] extracted SURF features of the pre-processed strip images and then performed block registration, which achieved good mosaic results. To obtain sonar images of large-area seafloor surface, Zhao et al. [16] also proposed a side-scan image mosaicking method based on the coupling feature points of position constraints. In addition, He et al. [17] used the unsharp masking (USM) algorithm to enhance the side-scan images and the SURF algorithm for image mosaicking; experiments showed that their method effectively enhanced image features and increased the amount of image information, but the average gray values of the images were affected.

The above image mosaic algorithms primarily focused on the extraction and registration of features points of adjacent strip images, and most adopted the wavelet fusion algorithm after image registration, without further exploration for alternative image fusion algorithms. Due to the complex marine environment during ocean exploration, it is nearly impossible to ensure that the sonar images on one survey line are always better than those of an adjacent strip image. Therefore, it is necessary to take into account the differences in image resolution during strip image mosaicking and retain clear image information while screening necessary information in blurred images. To address this problem, we performed image fusion using curvelet transform, which can reveal more detailed information of strip images than wavelet transform. Then, the resolution of strip images was evaluated using a resolution weight model to constrain the curvelet transform, thereby achieving mosaicked strip images with better quality. The contents of this paper were arranged as follows: Section 2 mainly introduces seven different methods of resolution assessment, which would all be used in the calculation of resolution weight model; Section 3 mainly introduces the specific process of strip Mosaic method proposed in this paper; Section 4 uses the measured data to verify the feasibility of this method; and Section 5 contains the summary and prospects.

## 2. Image Resolution Assessment Methods

As an important data source of seafloor geomorphology, the resolution of side-scan sonar images directly determines the accuracy of target identification and seafloor sediments classification. The assessment of image quality can be divided into two types: subjective assessment and objective assessment [18,19]. Subjective assessment is mainly performed by trained professionals, whereas objective assessment uses mathematical models to measure the image resolution based on different indices. Thus, it is imperative to develop an objective assessment method that is in consistency with subjective assessment. Currently, common objective assessment methods can be divided into three categories according to the degree of use of reference images, i.e., full-reference quality assessment, reduced-reference quality assessment, and no-reference quality assessment [20]. Since there is no original reference image for side-scan sonar images, the no-reference quality assessment method was adopted in this study.

Image resolution is one of the most important image quality evaluation indexes and is the most important image parameter of sonar image. Therefore, the resolution of image became the main research object. A total of seven resolution assessment methods from four aspects will be introduced in this section. As the more classical parameter indexes in the assessment method, they measure the sharpness of the image from different aspects. Additionally, they will all be used in the calculation of resolution vector in Section 3, making the evaluation result more accurate and perfect.

### 2.1. Assessment Method Based on Image Gradient

Image gradient reflects the marginal information of images. The greater the gradient value is, the sharper the image edge and the clearer the image will be. Common gradient functions for evaluating image resolution include the following three types [21].

#### 2.1.1. Energy Gradient Function

The energy gradient of an image is the quadratic sum of the difference in grayscale value of adjacent pixels in the horizontal and vertical direction. The summation of energy gradient values of all pixels in the image is then taken as the function value. The function is shown in Equation (1):(1)$$F_{EG}=\sum_{x}\sum_{y}\{{\left[f\left(x+1,y\right)-f\left(x,y\right)\right]}^{2}+{\left[f\left(x,y+1\right)-f\left(x,y\right)\right]}^{2}\}$$
where *x* and *y* are pixel coordinates, and f(x,y) is the grayscale value of the pixel.

#### 2.1.2. Brenner Gradient Function

Brenner gradient function is relatively the easiest gradient assessment function [22]. It calculates the quadratic sum of the grayscale difference of two adjacent pixels, meaning a small calculation amount. Yet, it is sensitive to noise. The function is shown in Equation (2):(2)$$F_{Brenner}=\sum_{x}\sum_{y}{\left[f\left(x+2,y\right)-f\left(x,y\right)\right]}^{2}$$

#### 2.1.3. Tenengrad Gradient Function

Krotkv et al. [23] used the Tenengrad gradient function as one of the assessment indexes of image resolution, the results of which were close to objective assessment results. In this method, the Sobel operator was first used to extract the horizontal and vertical gradient values of pixels, then the quadratic sum was compared with a threshold T. The gradient values of pixels greater than T were added to obtain the Tenengrad gradient function value. The function is shown in Equation (3):(3)$$F_{Tenengrad}=\sum_{x}\sum_{y}\left[G{\left(x,y\right)}^{2}\right]$$
where G(x,y) is the gradient calculated by the Sobel operator, as shown in Equation (4):(4)$$G\left(x,y\right)=\sqrt{G_{x}^{2}\left(x,y\right)+G_{y}^{2}\left(x,y\right)}$$
where $G_{x}\left(x,y\right)$ and $G_{y}\left(x,y\right)$ represent the horizontal and vertical gradient values, respectively.
(5)$$\begin{array}{l} G_{x}\left(x,y\right)=f\left(x,y\right)⊗g_{x}\\ G_{y}\left(x,y\right)=f\left(x,y\right)⊗g_{y} \end{array}$$
where ⊗ is the convolution operator, and $g_{x}$ and $g_{y}$ represent the horizontal and vertical templates of the Sobel operator, respectively:(6)$$\begin{array}{c} g_{x}=\left[\begin{array}{ccc} -1 & 0 & 1\\ -2 & 0 & 2\\ -1 & 0 & 1 \end{array}\right]\\ g_{y}=\left[\begin{array}{ccc} -1 & -2 & -1\\ 0 & 0 & 0\\ 1 & 2 & 1 \end{array}\right] \end{array}$$

### 2.2. Assessment Method Based on Image Transform Domain

It is generally believed that a clear image contains more high-frequency components than a blurry image. Thus, some studies have attempted to transform the image to the frequency domain to perform image quality assessment [24].

#### 2.2.1. Discrete Fourier Transform (DFT)

As the most basic time–frequency transformation methods, DFT is widely used in resolution assessment. Specifically, 2D DFT is first performed on the image, and then the zero-frequency component is shifted to the matrix center, such that the frequency diffuses from the center to the periphery and from low frequency to high frequency. The spectrum values of corresponding pixels are weighted based on the distance to the central pixel, and the resolution assessment value is the weighted average of the spectrum values of all pixels [25,26]. The function of DFT-based image resolution assessment is shown in Equation (7) [27]:(7)$$F_{DFT}=\frac{1}{M\times N}\sum_{\mu =0}^{M-1}\sum_{\nu =0}^{N-1}\sqrt{\mu^{2}+\nu^{2}}P(\mu ,\nu )$$
where *M* and *N* are the image dimensions, $\sqrt{\mu^{2}+\nu^{2}}$ represents the distance of a pixel to the central pixel, and P(μ,ν) is the spectrum value of a pixel after DFT.

#### 2.2.2. Discrete Cosine Transform (DCT)

DFT-based resolution assessment methods have high sensitivity; however, they are computationally more demanding than DCT-based methods. In comparison, DCT has a general orthogonal transform property, and the base vector of DCT matrix could describe image features very well [28,29]. Therefore, by replacing DFT with DCT, the transform coefficient is changed into a real number, which reduces the computation while still obtaining the distribution of image frequency. The resolution assessment function based on DCT is shown in Equation (8):(8)$$F_{DCT}=\frac{1}{M\times N}\sum_{\mu =0}^{M-1}\sum_{\nu =0}^{N-1}\left(\lambda +\varphi \right)\left|C(\lambda ,\varphi )\right|$$
where C(λ,φ) is the spectrum value of a pixel after DCT.

### 2.3. Assessment Method Based on Entropy Function

The entropy of an image is an important index to measure the richness of image information. Shannon believed that the greater the entropy value, the richer information the image contains. During image resolution assessment, the clearer the image is, the more abundant grayscale distribution it has, and thus, the greater the entropy value is [30]. The definition of entropy function is shown in Equation (9):(9)$$F_{entropy}=\sum_{i=0}^{255}-p(i)\log_{2}p(i)$$
where p(i) is the probability of occurrence of every grayscale value.

### 2.4. Assessment Method Based on Variance Function

The variance function can represent the dispersion degree of the image grayscales. The smaller the range of grayscale, the smaller the variance is and the blurrier the image is, and vice versa [31]. The definition of variance function is shown in Equation (10):(10)$$F_{Var}=\sum_{x}\sum_{y}\{{\left[f(x,y)-\varepsilon \right]}^{2}\}$$
where ε is the average grayscale value of the image, the definition of which is in Equation (11):(11)$$\varepsilon =\frac{1}{M\times N}\sum_{x}\sum_{y}f(x,y)$$

## 3. Strip Mosaic Method Based on Curvelet Transform and Resolution Constraints

### 3.1. Image Fusion Algorithm Based on Curvelet Transform

To obtain a clear and continuous image that can reflect complete information of the entire testing zone, image fusion in the overlapping area of side-scan sonar strip images is required. Currently, there are three common image fusion methods, namely weighted average method, image pyramid method, and wavelet fusion method [32]. The wavelet fusion method is the most common side-scan sonar strip image mosaicking method. However, due to the limitations in algorithms, the wavelet transform can only obtain edge features in the horizontal and vertical directions, and the wavelet basis does not have the anisotropy property. Hence, it is unable to get close to the image texture features. To overcome the limitations in the wavelet transform and improve the quality of strip image mosaicking, the Curvelet transform was introduced in the current study.

The Curvetlet transform was first proposed by Candes and Donoho in 1999 [33] based on the Ridgelet transform. As a multi-resolution, band-pass, and directional multi-scale image analysis method, Curvelet transform has the three characteristics of an optimal image representation method proposed by the National Institute for Physiological Science, Japan [34]. Similar to wavelet transform, Curvelet transform calculates the correlation of spatial images using a group of base functions, thereby characterizing edges and curves at different angles. The main steps of image fusion based on Curvelet transform are as follows: Curvelet coefficients are first obtained from Curvelet decomposition of the image, the coefficients are then processed based on specific fusion rules, and lastly, inverse Curvelet transform is carried out on the fused coefficient to obtain the final fusion image [35,36].

The Curvelet coefficients are obtained using the equation below:(12)$$C\left(j,\theta ,k_{1},k_{2}\right)=\sum_{0\leq x\leq M,0\leq y\leq N}f(x,y)\cdot \varphi_{j,\theta ,k_{1},k_{2}}(x,y)$$
where f(x,y) is the input image, M×N are the image dimensions, j is the scale, θ is the direction, $k_{1},k_{2}$ is the spatial location of Curvelet, and φ(x,y) represents the Curvelet function, which includes a group of base functions described by parameters $\left(j,\theta ,k_{1},k_{2}\right)$.

Different from the wavelet coefficients, the Curvelet coefficients include the low-frequency coefficient in the innermost layer (i.e., the Coarse layer), the mid-to-high frequency coefficient in the Detail layer, and the high-frequency coefficient in the outermost Fine layer. As the number of layers increases, the scale of the corresponding base function turns smaller, and there are more directions. Figure 1 shows a frequency-domain base division method. Each square in Figure 1 represents a scale, and there are five scales. The bigger the square, the higher the frequency, and the smaller the scale is; hence, more detailed information will be reflected. The radial lines represent the angles. At each scale, the angle division is different, and the higher the frequency is, the smaller the angle is.

From Jia et al. [37], the energy of coefficients is mainly concentrated in the low-frequency coefficient, and the energy gradually declines as the frequency increases. In other words, the low-frequency coefficient reflects the general trend of the image, whereas high-frequency coefficient reflects the outline and texture details of an image. By fusing the coefficients at various layers using different fusion rules, the fusion image coefficient can be obtained, and by performing inverse Curvelet transform of the fusion image coefficient, the fusion image is obtained.

### 3.2. Strip Image Mosaicking Based on Curvelet Transform and Resolution Constraints

Due to uncertainties in the marine environment during exploration, common areas in adjacent strip images might have large differences during actual measurement. Both strip images might have good quality, or one or both of them may not be good at all. The traditional side-scan strip image mosaicking algorithms do not take the image resolution into account. In order to ensure good mosaic results, a Curvelet coefficient fusion criterion based on the resolution weight model was proposed in the present study.

In Section 2, we have introduced seven different image resolution assessment methods, including energy gradient function, Brenner gradient function, Tenengrad gradient function, DFT, DCT, entropy function, and variance function. According to Li et al. [38] and Xie et al. [39], different resolution assessment methods may have different results for the same group of images. In other words, a single method is not able to assess the resolution of an image accurately. Hence, these seven resolution assessment methods were integrated in this study to build a resolution vector, and the image resolution was obtained based on probability and given weights.

The resolution vector Q, created based on the resolution value of the above seven methods, is shown in Equation (13):(13)$$Q=[F_{EG},F_{Brenner},F_{Tenengrad},F_{DFT},F_{DCT},F_{entropy},F_{Var}]$$

Since the resolution index in each method has a positive relationship with the image resolution, the resolution weight is obtained by comparing the resolution vectors of image 1 and image 2, $Q_{1},Q_{2}$, respectively.
(14)$$Ratio=\frac{sum(Q_{1}\geq Q_{2})}{7}$$
where the resolution weight Ratio represents the probability of an image having better resolution than the other image. Thus, it was taken as the fusion rule in the Coarse layer of Curvelet transform, as shown in Equation (15).
(15)$$C_{Coarse\_fusion}=Ratio\cdot C_{Coarse\_1}+(1-Ratio)\cdot C_{Coarse\_2}$$
where $C_{C\mathrm{oarse}\_fusion}$, $C_{C\mathrm{oarse}\_1}$ and $C_{C\mathrm{oarse}\_2}$ represent the coefficient in the Coarse layer after fusion and that of image 1 and image 2, respectively.

In order to fully show the texture and details of the image, the maximum coefficient fusion approach was adopted to process the Detail layer and Fine layer coefficients, as shown in Equation (16):(16)$$\begin{array}{c} C_{Detail\_fusion}(x,y)=Max\{\left|C_{Detail\_1}(x,y)\right|,\left|C_{Detail\_2}(x,y)\right|\}\\ C_{Fine\_fusion}(x,y)=Max\{\left|C_{Fine\_1}(x,y)\right|,\left|C_{Fine\_2}(x,y)\right|\} \end{array}$$
where $C_{Detail\_fusion}$, $C_{Detail\_1}$ and $C_{Detail\_2}$ represent the coefficient in the Detail layer after fusion and that of image 1 and image 2, respectively. $C_{Fine\_fusion}$, $C_{Fine\_1}$ and $C_{Fine\_2}$ represent the coefficient of the Fine layer after fusion and that of image 1 and image 2, respectively. Figure 2 shows the flowchart of the proposed mosaic method based on Curvelet transform and resolution constraints.

Extract and match feature points of adjacent strip images and obtain registered mosaic strips using the affine transformation.Select the common area A from two strip images.Perform Curvelet transform for two images to obtain the coefficients in the Coarse layer, Detail layer, and Fine layer.Calculate the resolution vectors of the two images to obtain the corresponding resolution weight.Fuse the Coarse layer coefficients using resolution fusion rules to obtain the low-frequency coefficients. Fuse the Detail layer and Fine layer coefficients using the maximum coefficient fusion rules to obtain the high-frequency coefficients.Perform inverse Curvelet transform on the fusion coefficients to obtain the fusion image in area A, which is then mosaicked to the registered strip images.Repeat steps 2–6 until the whole mosaic image is obtained.

In traditional mosaic algorithms for strip images, there are various problems, such as inconsistent resolution of adjacent strip images and image distortion. In this study, we proposed a mosaic method for strip images based on Curvelet transform and resolution constraints, which produced mosaic images with complete information and high quality.

## 4. Experiment and Results

To verify the effectiveness of the proposed image mosaicking method, image data collected in 2019 using the Klein4000 side-scan sonar in Jiaozhou Bay, Qingdao, Shandong Province, China was used in the experiment. The water depth of the survey area is approximately 30–40 m. The overlapping rate of adjacent strip images is 50%. After preprocessing, such as seafloor tracking, slant-range correction, gray level equalization, noise suppression, gain correction, and geocoding, a group of strip image pairs with obvious common features were selected, as shown in Figure 3.

Figure 4a shows a mosaic image calculated based on geographic coordinate information. As can be seen, there is obvious dislocation and distortion. According to the steps of our method, the feature points in the strip images were extracted and matched, as shown in Figure 4b. Figure 4c shows a registered strip image after affine transformation. Based on the results, the distortion and dislocation were eliminated after image registration, resulting in good visual effects and laying a solid foundation for image fusion in the next step.

To effectively select the fusion area and ensure the integrity of the selected features, the whole survey area was first rotated counterclockwise for a certain angle, such that the survey line was approximately along the vertical direction [40]. Another reason to rotate the strips is that a series of subsequent steps, such as Curvelet transform and image fusion, require regular rectangles. After image mosaicking, it was rotated back to the original direction. Areas 1–3 were selected, and the sonar images of two strips in these areas are shown in Figure 5.

Taking Area 1 as an example, the proposed algorithm was used to process two strips in the area. First, the coefficients in the Coarse, Detail, and Fine layers were extracted using Curvelet transform. The coefficient structure is shown in Table 1.

In both strips, Area 1 has the same dimensions of 923×166. Five layer decomposition was carried out. As shown in Table 1, the dimension of the coefficient matrix increases with the increase in scale. The larger the scale in spatial domain, the smaller the scale in frequency domain, and the more detailed the description of high frequency information.

Then, the resolution vectors of two strips in Area 1 were calculated, and the results are shown in Table 2.

Then, using the proposed algorithm, the coefficients in the Coarse layer of the two images were fused based on the resolution fusion rule, and the coefficients in the Detail and Fine layers of the two images were fused using the maximum coefficient fusion approach, thereby obtaining the low-frequency and high-frequency coefficients of the fused image. Lastly, the fused image of Area 1 was obtained via inverse Curvelet transform.

In order to verify the rationality of the resolution fusion rule proposed in this paper, the resolution fusion rule, the mean fusion rule, and the maximum fusion rule are used to combine the five layer coefficients of the two images obtained by the Curvelet decomposition, respectively. As shown in Figure 6, 19 combinations of fusion coefficients were obtained.

Then, the fusion coefficients of each group were inversely transformed to obtain fusion images.

The information entropy, average gradient, and spatial frequency were used as evaluation indices of the fusion results. The information entropy reveals the amount of information contained in the image, and the greater the entropy, the better the fusion result; the average gradient reflects the image’s contrast expression of small details, and the greater the average gradient, the higher the image fusion quality; the spatial frequency represents the overall activity of the image in spatial domain, and the higher the spatial frequency, the better the fusion result. Table 3 shows the three indices of each combination, and Figure 7 shows the line chart of the analysis results.

As shown in Table 2 and Figure 7, the Curvelet coefficient fusion strategy proposed in this paper, namely the resolution fusion rule used in the Coarse layer and the maximum coefficient fusion rule used in the Detail layer and Fine layer, has the best image fusion effect.

To further demonstrate the effectiveness of the proposed algorithm, the images were fused using different algorithms, including simple average, traditional wavelet fusion and wavelet fusion with resolution constraints. The fusion results were compared with that of the proposed algorithm. The traditional wavelet fusion algorithm applies the mean fusion rule to the low-frequency information of wavelet transform and the maximum coefficient fusion rule to the high-frequency information. In the wavelet fusion with resolution constraints, the resolution fusion rule is applied to the low-frequency information of wavelet transform and the maximum coefficient fusion rule is applied to the high-frequency information.

Table 4 shows the three indices of the four fusion methods, and Figure 8 shows the fusion strip images.

As shown in Table 4, the information entropy, average gradient, and spatial frequency of the proposed algorithm are much greater than those of the other three methods, indicating that the fusion result of the proposed method is the best. By comparing the results of wavelet fusion with resolution constraints and our method, it can be seen that Curvelet fusion achieved better fusion results than wavelet fusion. In addition, based on the value of indices of traditional wavelet fusion and wavelet fusion with resolution constraints, the effectiveness of the resolution fusion rule proposed in this study was demonstrated. It can also be seen intuitively from Figure 8 that the fusion image obtained by our method has better clarity and can show more details.

To further verify the effectiveness of the proposed method, the same experiments were repeated for Areas 2 and 3. Figure 9 shows the fusion strip images in Area 2 and Area 3. The evaluation results are shown in Table 5.

As shown in Table 5, the fused images in Areas 2 and 3 of the proposed method have the highest information entropy, average gradient, and spatial frequency, suggesting the best performance in image fusion and validating the effectiveness and stability of the proposed algorithm.

Then, the fused images in the three areas were mosaicked onto the registered strip, which was then rotated clockwise to the original orientation, as shown in Figure 10. Compared with Figure 4c, it can be seen that Figure 9 better reflects the overall characteristics of the features by enhancing detail texture information while retaining the overall trend of the overlapping areas.

## 5. Conclusions

Current strip image mosaicking algorithms do not consider the influence of the resolution difference of common objects in adjacent images on the results of mosaicking. Moreover, a traditional wavelet fusion algorithm is not able to fully describe the image details. To address these problems, in this study, we proposed an image mosaic method based on Curvelet transform and resolution constraints. Experimental verification using actual measurement data showed that the proposed method can greatly improve the fusion results, which provides high-quality image data for subsequent submarine target recognition and sediment classification, thereby greatly benefiting ocean exploration. However, there are still a lot of improvements to be made in this method, such as human involvement in the process. In view of this, target recognition and other technologies in deep learning can be introduced in the future. Thus, it can automatically identify and extract the areas that need to be fused and achieve full automation.

## Figures and Tables

**Figure 1 sensors-21-06044-f001:**
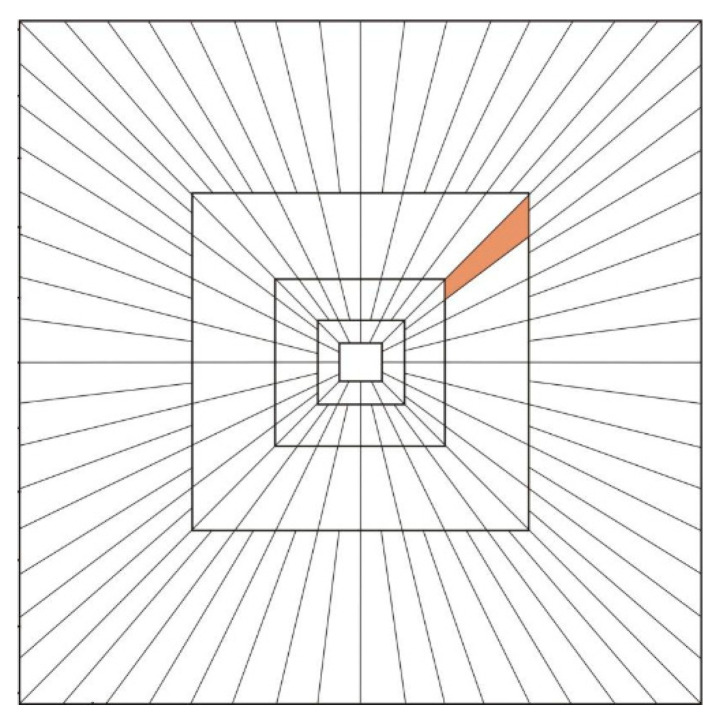
Frequency-Domain Base of Curvelet Transform.

**Figure 2 sensors-21-06044-f002:**
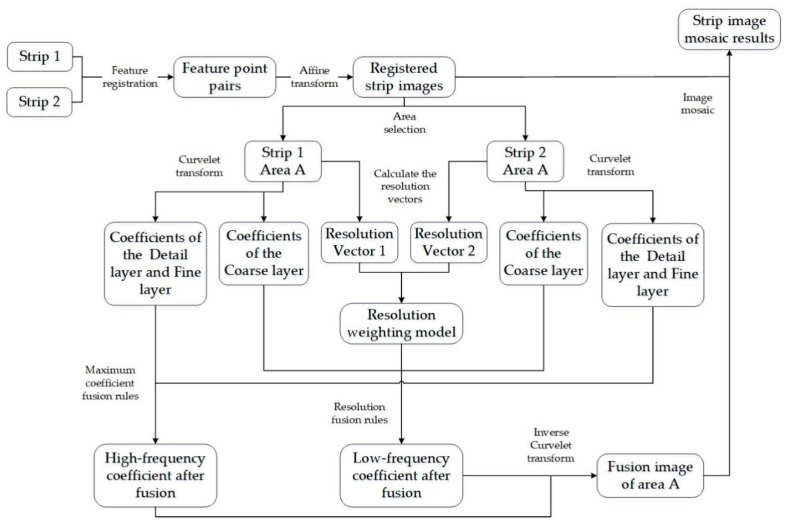
Flowchart of the proposed method.

**Figure 3 sensors-21-06044-f003:**
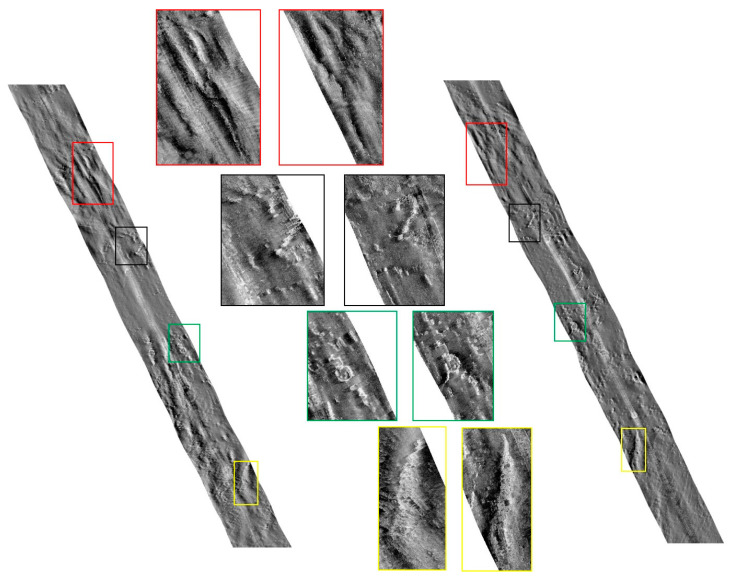
Two strips used for verification. Four image pairs with obvious common features were selected.

**Figure 4 sensors-21-06044-f004:**
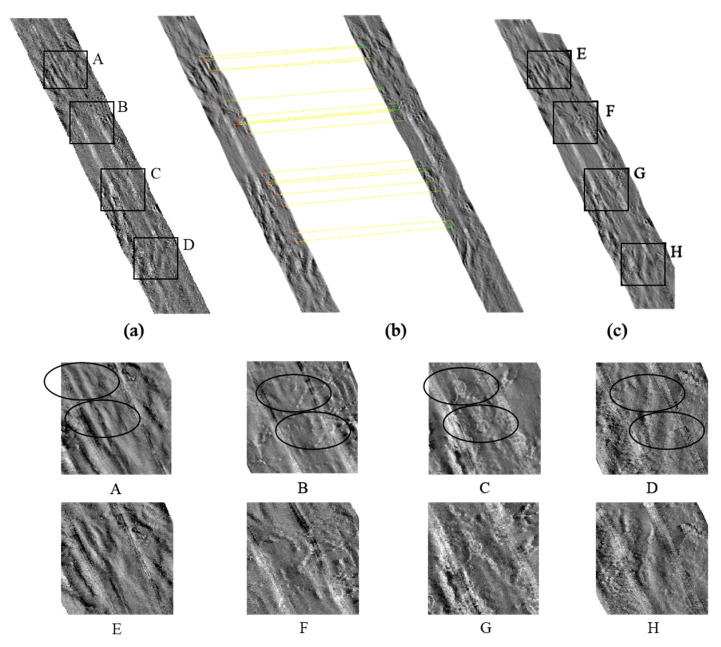
Strip image registration. A–D and E–H are four areas selected from (**a**) and (**c**) respectively. (**b**) shows the registration process of strips. It can be seen that there was significant dislocation in A–D. After strip registration, the dislocation effect largely disappeared in E–H.

**Figure 5 sensors-21-06044-f005:**
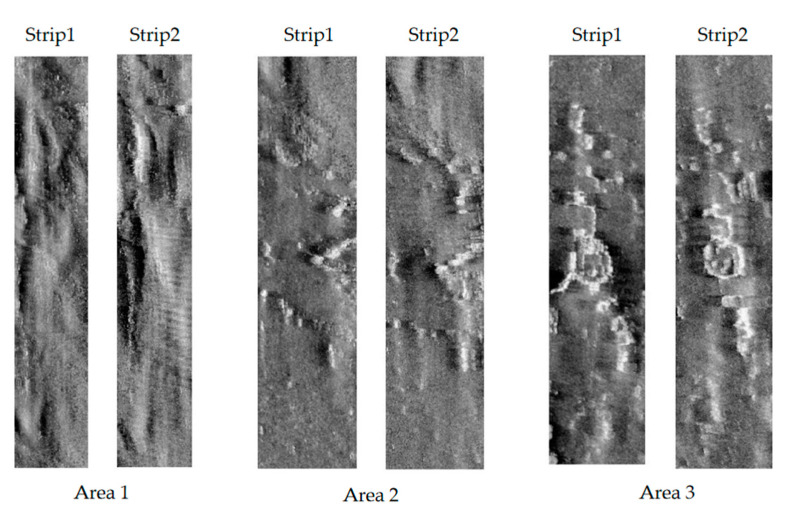
Selected fusion areas.

**Figure 6 sensors-21-06044-f006:**
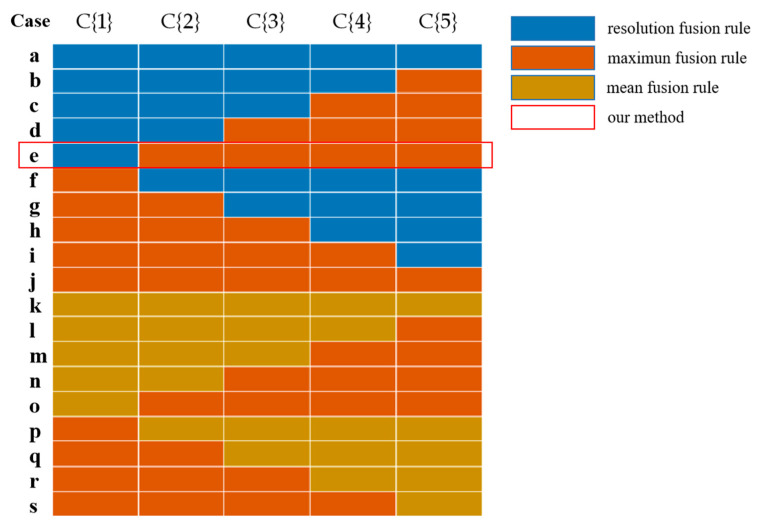
Combination diagram of fusion rules about Curvelet transform coefficients.

**Figure 7 sensors-21-06044-f007:**
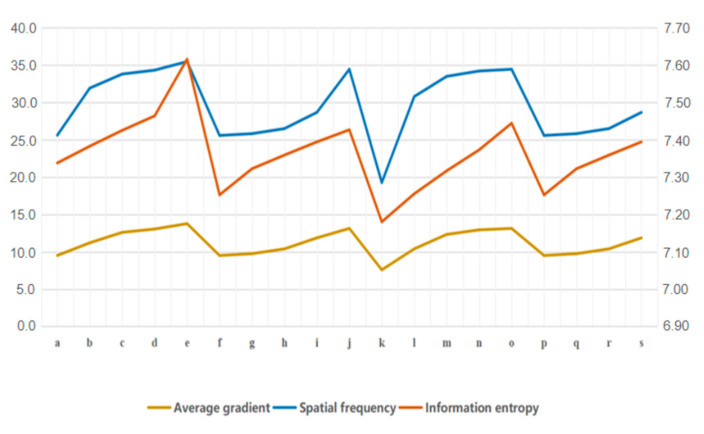
Line chart of the fusion coefficients analysis results.

**Figure 8 sensors-21-06044-f008:**
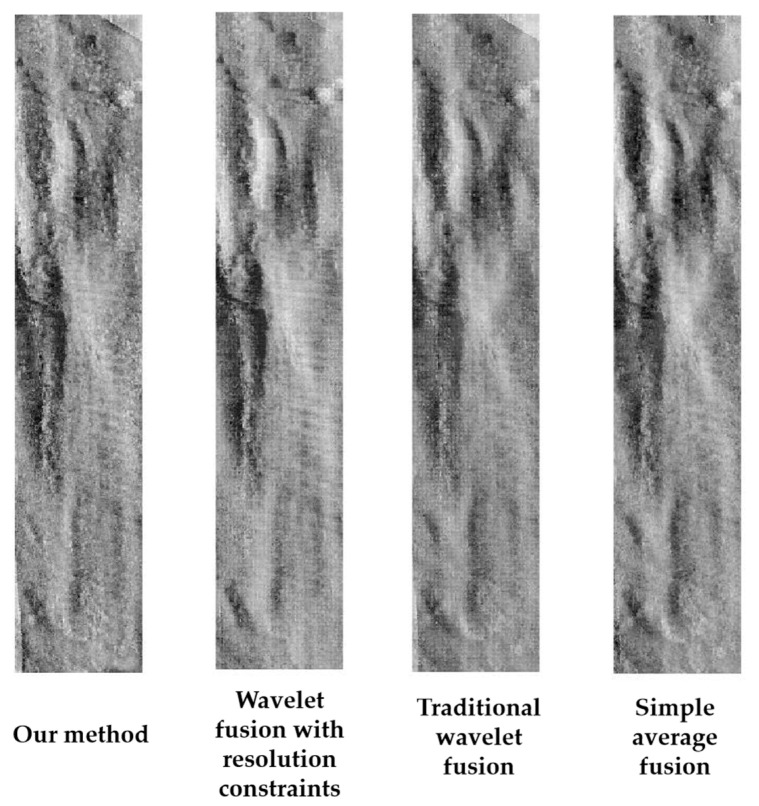
Fusion results of the four different methods.

**Figure 9 sensors-21-06044-f009:**
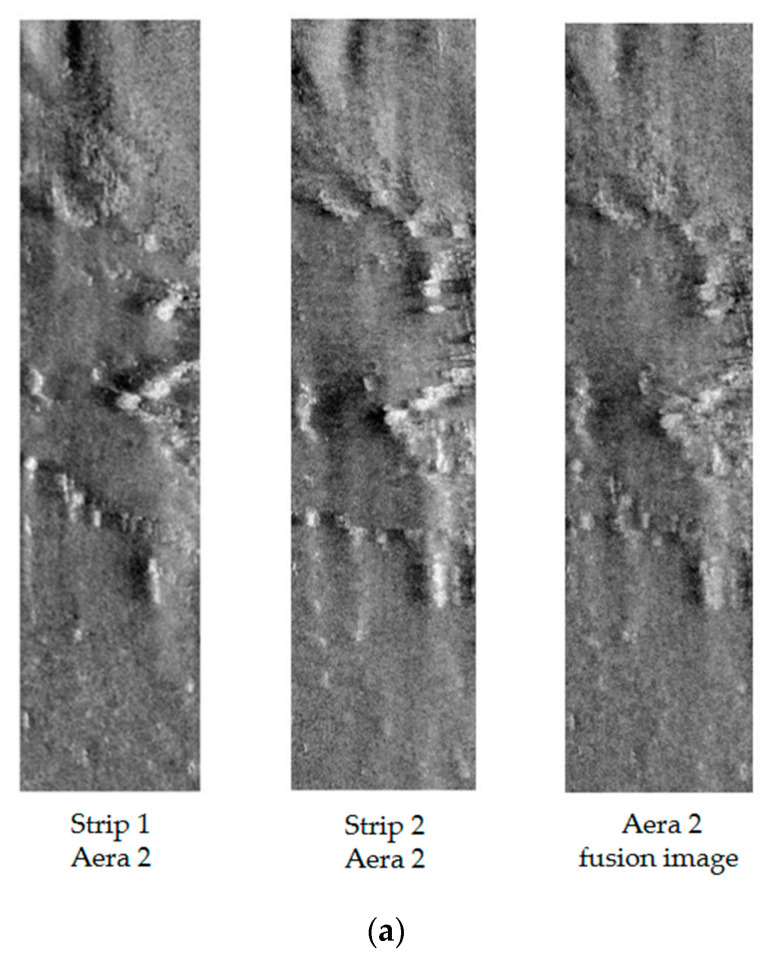
(**a**) shows the strip images and fusion strip images in Area 2. (**b**) shows the strip images and fusion strip images in Area 3.

**Figure 10 sensors-21-06044-f010:**
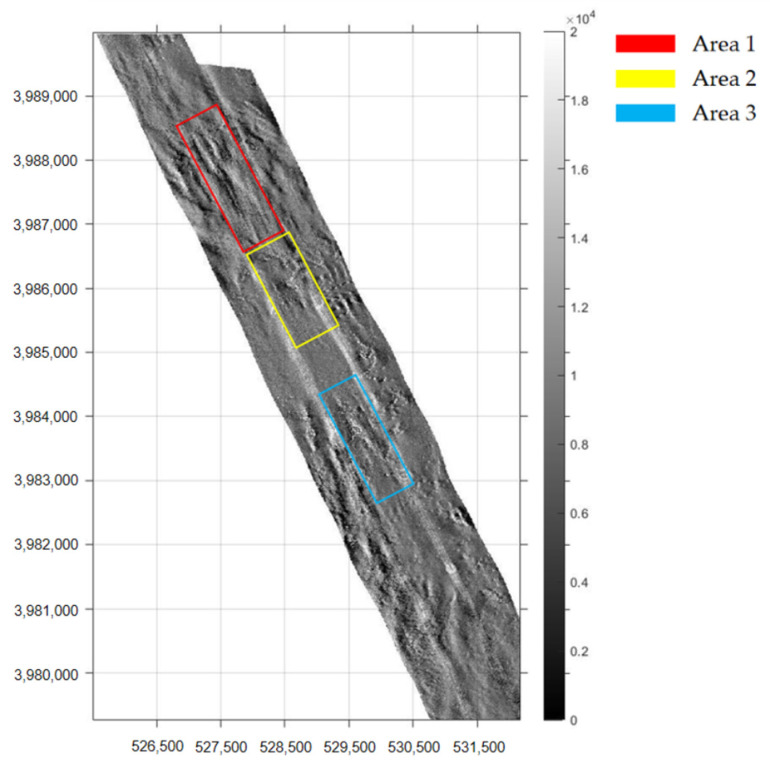
Results of strip image mosaicking.

**Table 1 sensors-21-06044-t001:** Structure of Curvelet transform coefficients.

Layer	Scale Coefficient	Number of Directions	Matrix Dimensions
Coarse	C{1}	1	77×13
Detail	C{2}	16	62×14 57×14 77×11 77×10
	C{3}	32	120×14 115×15 115×14 115×15 77×22 78×21 77×21
	C{4}	32	241×29 231×28 231×29 154×44 155×42 155×42 154×42
Fine	C{5}	1	923×166

**Table 2 sensors-21-06044-t002:** Resolution vectors of two strips in Area 1.

	$F_{EG}$	$F_{\mathrm{Brenner}}$	$F_{\mathrm{Tenengrad}}$	$F_{DFT}$	$F_{DCT}$	$F_{\mathrm{entropy}}$	$F_{\mathrm{Var}}$
$Q_{1}$	$1.6\times 10^{9}$	$7.2\times 10^{8}$	$6.0\times 10^{8}$	$2.5\times 10^{8}$	$4.3\times 10^{3}$	$2.6\times 10^{6}$	7.303
$Q_{2}$	$2.2\times 10^{9}$	$1.4\times 10^{8}$	$1.0\times 10^{8}$	$3.4\times 10^{8}$	$6.2\times 10^{3}$	$3.3\times 10^{6}$	6.886

$Q_{1},\;Q_{2}$ denote the resolution vectors of Strip 1 and Strip 2, respectively. Additionally, the resolution weight ratio, computed according to Equation (14), is 0.1428.

**Table 3 sensors-21-06044-t003:** Comparison of fusion effects in different combinations.

Case	Information Entropy	Average Gradient	Spatial Frequency
a	7.3376	9.4983	25.5944
b	7.3826	11.1938	31.9010
c	7.4250	12.5927	33.7882
d	7.4637	13.0406	34.3080
e ^1^	7.6156	13.7586	35.4629
f	7.2523	9.4976	25.5687
g	7.3222	9.7393	25.8062
h	7.3588	10.3818	26.4863
i	7.3941	11.8541	28.6577
j	7.4265	13.1232	34.4313
k	7.1802	7.5718	19.2626
l	7.2552	10.3865	30.7766
m	7.3167	12.3194	33.4798
n	7.3728	12.9310	34.1983
o	7.4441	13.1217	34.4303
p	7.2523	9.4976	25.5687
q	7.3222	9.7393	25.8062
r	7.3588	10.3818	26.4863
s	7.3941	11.8541	28.6577

^1^ is the fusion rule combination form of our method.

**Table 4 sensors-21-06044-t004:** Comparison of fusion results of different methods in Area 1.

Algorithms	Information Entropy	Average Gradient	Spatial Frequency
Our method	7.6156	13.7586	35.4629
Wavelet fusion with resolution constraints	7.3569	9.0872	28.6397
Traditional wavelet fusion	7.2260	8.2050	26.8381
Simple average	7.1584	7.6452	19.5543

**Table 5 sensors-21-06044-t005:** Comparison of fusion results of different methods in Areas 2 and 3.

	Ratio	Fusion Algorithms	Information Entropy	Average Gradient	Spatial Frequency
Area 2	0.1428	Our method	7.1318	11.7527	30.4386
		Wavelet fusion with resolution constraints	6.9388	8.2569	25.4472
		Traditional wavelet fusion	6.7614	7.3451	23.7968
		Simple average	6.6962	6.7457	17.0373
Area 3	0.2857	Our method	7.2367	11.8425	30.1219
		Wavelet fusion with resolution constraints	6.9619	7.8150	24.4447
		Traditional wavelet fusion	6.9174	7.4510	23.7001
		Simple average	6.8657	6.8585	16.9889

## Data Availability

The data is kept in_house and not available online.

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
