# Peer review of "A Mosaic Method for Side-Scan Sonar Strip Images Based on Curvelet Transform and Resolution Constraints"

_sensors, 2021, doi:10.3390/s21186044_

Round 1

Reviewer 1 Report

This paper describes a computational framework for mosaicking of side-scan sonar strip images based on curvelet transform and resolution constraints, reducing the influence of resolution difference of overlapping areas on two adjacent strip images. In general, the paper is clearly presented and easy to follow. Additionally, the topic is useful and fits the scope of the Sensors journal.

The major concern is that experimental results are not sufficient. The authors only show results on limited sonar strip images with simple backgrounds and ample overlapping areas. Please use more challenging images. Moreover, the proposed method is only compared with a number of basic fusion/ mosaicking methods based on wavelet decomposition or simple averaging. It is difficult to evaluate the benefits of the proposed method. Please involve more recently proposed methods and provide more solid/convincing evaluation results.

[1] Zhao, Jianhu, Xiaodong Shang, and Hongmei Zhang. "Side-scan sonar image mosaic using couple feature points with constraint of track line positions." Remote Sensing 10, no. 6 (2018): 953.

[2] Tang, Zhijie, Zhihang Luo, Lizhou Jiang, and Gaoqian Ma. "A novel high precision mosaic method for sonar video sequence." Multimedia Tools and Applications 80, no. 9 (2021): 14429-14458.

[3] Zhao, Jianhu, Aixue Wang, Hongmei Zhang, and Xiao Wang. "Mosaic method of side-scan sonar strip images using corresponding features." IET Image Processing 7, no. 6 (2013): 616-623.

[4] Tang, Zhijie, Gaoqian Ma, Jiaqi Lu, Zhen Wang, Bin Fu, and Yijie Wang. "Sonar image mosaic based on a new feature matching method." IET Image Processing 14, no. 10 (2020): 2149-2155.

Reviewer 2 Report

This paper describes a method for fusing overlapping images in a mosaic of marine side-scan sonar data.  The method appears to be a fairly logical evolution of methods already described in the literature, and amounts to a new way of weighting information from the two images in which different scales (in a curvelet transformation) are treated differently based on estimates of the relative resolution of the images.  The method is demonstrated on one pair of images with same-side illumination (with three subareas examined in detail) and appears to give useful results.  Objective measures of the quality of the merged data are more favorable for the new method, but to my subjective eye the results are different in the appearance of individual features but comparable in quality to existing methods. Thus, this work is evolutionary, but it is clearly relevant to Sensors and may be of interest to some readers.  Because I think the work is valid, I recommend that it be published after minor revisions to the text as described below.  The main intent of these suggestions is for the authors explain more clearly how their approach relates to past methods and why they made the specific modifications they describe.  A broader perspective on how the method is likely to work for other images would also improve the paper.  I am not asking for changes to the to the research, but, having said that, if the authors could substitute a different pair of sonograms with a more dramatic difference in resolution (so that the merits of their and other fusion approaches would be more obvious to the eye) I think this would improve the paper immensely.  Using a more dramatic example case would be the greatest single improvement the authors could make, but I will leave this as optional for them to consider.

Overall, the paper is well organized and complete.  The writing would benefit from light copyediting (e.g., to ensure a consistent tense and to eliminate typographical errors) but is generally very clear.  The notation used is clear and properly explained.  Equations, figures, and tables are clear and contribute materially to the reader’s understanding of the results.

The most important topic on which I’d like to see additional discussion added to the paper is the relation of the new method that is being tested to the existing curvelet (and wavelet) fusion approaches.  The authors do survey the history of sonar image mosaicking and document that this series of methods has been proposed.  What I’d like to see is more discussion of (a) the reasons for using curvelets rather than wavelets, and (b) the reasons for the specific choices for blending curvelet coefficients that they have made (including stating explicitly what is done in the “traditional curvelet fusion”).

In terms of the shortcomings of the wavelet approach, I am puzzled by the statement (lines 192-194) that “the wavelet transform can only obtain edge features in the horizontal and vertical directions” which is taken to be a shortcoming because “it is unable to get close to the image texture features.”   It may be true that the wavelet basis does not have the property of anisotropy, but surely it is a complete basis, which means it can in principle describe any content that may occur in the images. (The bases for FFT, DCT, wavelets, and presumably curvelets too, are complete and thus capable of representing arbitrary images.  Where the differ is how accurately the transform represents the image when all but the most important coefficients are set to zero.  This property is relevant to data compression but not to the fusion application in this paper.)  The curvelet transform does explicitly bring orientation into the description of the basis functions, but is this appropriate?  The Wikipedia article on curvelets (https://en.wikipedia.org/wiki/Curvelet) states

“Curvelets are an appropriate basis for representing images … which are smooth apart from singularities along smooth curves, where the curves have bounded curvature, i.e. where objects in the image have a minimum length scale. This property holds for cartoons, geometrical diagrams, and text. … However, natural images (photographs) do not have this property; they have detail at every scale. Therefore, for natural images, it is preferable to use some sort of directional wavelet transform whose wavelets have the same aspect ratio at every scale.” [emphasis in the original]

Sonar images would seem to fall in the class of “natural images” rather than drawings, so why are curvelets considered appropriate at all?  Note that I am _not_ saying the authors are incorrect in their facts or in choosing wavelets.  Instead, I am saying that to a reader like myself with only a limited knowledge of curvelets, the reasons for using them in this context are not at all clear and need to be explained more carefully. This explanation can probably be provided by citing and quoting the existing literature in slightly more detail.

Next, as pointed out in the paper (lines 231-233), there can be many curvelet fusion methods that differ in how the curvelet coefficients are combined.  The specific choices the authors made are explained clearly so that readers could attempt to duplicate them. The differences from past methods and the reasons for choosing these specific new rules are not explained, and they should be.  What are the “traditional” rules for fusing curvelet coefficients? What are the shortcomings of the results with the “traditional” method, and do the new rules have a clear logic that relates them to fixing these shortcomings?  Or are the changes empirical?  Did the authors experiment with different sets of rules for mixing the coefficients and choose the best result?  (An obvious example of an alternative rule would be to apply the weighting of Eq. 15 to some or all of the detail layers rather than using the maximum coefficient in these layers per Eq. 16.  Another fairly obvious idea would be to weight the coefficients being fused according to how _much_ one image is sharper than the other, rather than merely according to how many of the resolution measures say it is the better of the two.  There are many other possibilities.)  In other words, how did the authors go about picking a “better” rule and do they have grounds for thinking it is the best choice, at least from some subset of possible fusion rules?

A second general area of discussion that I would like to see expanded is the likely range of applicability of the new fusion approach.  The paper mentions the possibility of needing to blend good with good, good with bad, or bad with bad quality images to make a mosaic. The single example strikes me as good with good:  the two images are illuminated from the same direction, their resolution is (to my eye) rather similar, and both have a fairly low noise level and clear surface features.  Thus, a negative aspect of the applicability of the method is how it performs in more challenging cases. Before adopting the new fusion approach, a user would probably want to know how it will perform on noisy images, images with greater resolution disparity, noisy images, images with opposite illumination, and even in cases where the registration process did not work well so the features are not precisely aligned.  The robustness of a method to such problems is as important as the method’s peak performance on favorable data.  Do the authors have any other results that show how their approach behaves with different data?  (My suggestion, not a requirement, to use a pair of images with a more visible difference in resolution would certainly fit well as the first step toward showing how the method works on multiple datasets with different quality issues.  However, the main point of that suggestion was to make the advantages of the proposed method more visible to the reader rather than to prove wider applicability.)  If not, can the authors make any theoretical arguments about the robustness of their method compared to the others?

There is also a positive side to applicability.  Do the authors think the method could be useful for types of data other than side-scan sonar, such as imaging radar, optical and infrared images, or even biomedical images?  If so, describing other ways the method might be useful would increase the impact of the paper.

Minor comments, in order of occurrence

L 8-9 I think the first sentence of the abstract, “Due to complex marine environment, the side-scan sonar signals are unstable, resulting in random non-rigid distortion in the side-scan sonar strip images” starts the reader off in the wrong direction, even though it is quite true.  The paper is not about the need to remove distortions and register images.  It is about the need to mosaic multiple images and thus to do some kind of blending of the seams to achieve a good appearance.  In the tests of blending algorithms, the images are first registered, but this is taken as a given step and not described in great detail.  I think this is totally appropriate.  I would suggest removing the first sentence about registration from the abstract. In Section 1 they can make it clear that registration is a challenge but that the paper is about the challenge that comes after registration, the fusion step.

L 10 “Public” areas should probably be “common” areas.  I was concerned when I read this poor choice of English word for the intended concept that the paper would contain other errors of translation.  I am glad to say that this is the only problem with word choice that I noticed.

L 31-53 These lines are all about the challenge of correctly registering sonar data, which is not the focus of the paper, so perhaps this amount of detail is not needed.  It would suffice to indicate that this step is hard but has been studied by many authors, and indicate which of the methods that are currently listed is going to be used to register the test data for this paper.

L 62-64  I have a number of comments about this and the following lines.  First, this sentence talks about one image being “better than” the adjacent one.  “Better” could encompass more than one criterion, such as signal to noise ratio or minimizing shadows in addition to resolution, but the rest of the paper focuses entirely on differences in resolution.  Perhaps a transitional sentence is needed, saying that the paper focuses on resolution differences because this is the most important way that sonograms differ in quality.  Second, “the complex marine environment” is unsatisfyingly vague as an explanation for differences in image quality.  Can the authors describe the most important effects.  For example, is range a determining factor, such that the near edge of an image is always higher quality than the far edge?  Do effects that vary in time and place, such as impurities in the water, play a role?  These would indeed be hard to predict. The same is true for wave activity. Does it affect image quality by causing the sonar to move less predictably?

Part of the reason for these questions is because it was not clear to me whether the relative quality and thus fusion weighting can be determined once for the whole overlap area between two images, or whether quality varies and the fusion weighting has to be estimated for local areas.

L 83  It would also be very useful to discuss in the introductory part of Section 2 how these “sharpness” measures are sensitive to effects other than image sharpness.  Increased gradients can indicate increased sharpness, but also an increase in noise or simply a contrast stretch of the whole image.  Noise and overall contrast also affect the entropy and variance.  Are these measures different enough from one another that they make up for each other’s shortcomings?  I see some advantage to having multiple types of gradient estimator, because they include different amounts of implicit smoothing of the image.  Thus, they differ in sensitivity to pixel-by-pixel noise.  (An interesting line of research would be to add other resolution measures that include more explicit control of the amount of smoothing, such as Laplacian-of-Gaussian filters.)  In general, though it seems that all the statistics will say an image is “sharper” if it has merely been stretched to increase its contrast, so this strikes me as a weakness.

L 71 ff Section 2 on methods to assess resolution is generally quite clear in its content, but its position in the paper is less clear.  I found it a little confusing because the section describes several methods for estimating sharpness but ends without saying how these methods are going to be used.  To help the reader understand the flow of the paper, the authors could add a few sentences describing how the sections relate to one another.  This could be a brief outline of the rest of the paper at the end of the Introduction, or it could just be a sentence at the beginning of section 2 saying something like “Our fusion approach depends on identifying the image of the overlapping pair that has the higher resolution, so we begin by discussing different methods of quantifying image resolution.”  It would also be possible to describe the curvelet transform and blending of curvelet coefficients (Section 3) first and include a sentence that says something like “The seven resolution measures that we use in our resolution vector are described and equations for them are given in the next section.”  The current Section 2 could then be placed after Section 3.  Of course the two ideas (moving Section 2 and providing an outline of the paper) could be combined.

L 150 Section 2.2.1 on DFT is not entirely satisfactory at present.  Equation 7 gives the definition of the resolution measure in terms of the power spectrum P(mu,nu) but the DFT itself and how to compute the power spectrum are not specified.  This material could be added, but I think it suffices to cite a reference that gives these definitions at the end of the section.  Note that references [25, 26] appear from their titles to be specifically about the sharpness estimate.  If one or both of these papers explicitly describe how to compute the power spectrum, then it would be helpful to the reader to say this explicitly, e.g., “The power spectrum P(mu, nu) is computed as described in [25].  Alternatively, a general work on use of the DFT with the appropriate equations could be cited.

L 162 The exact same remarks apply for the DCT as for the DFT.  Section 2.2.2 should include a reference that provides the information needed to compute the DCT-based spectrum C.

L 199-200 Surely a reference should be cited for “the three characteristics of an optimal  image  representation  method  proposed  by  the  National  Institute  for  Physiological Science, Japan.”

L 300 ff I just want to say that Section 3 is especially well organized and the presentation of the test is well thought out and effective (apart from the quality difference between the two images  being so subtle).

L 312-318  This paragraph and Figure 4 are OK but I think it is not a good description to say that the correspondences in  Fig. 4(b) and rectification in Fig. 4(c) were performed “using the proposed method.”  This paper is about a proposed (new) method for image fusion at seams.  It makes use of an existing method for registering the images.  It does not “propose” this registration method, or describe it in great detail, or test it other than to confirm with Figure 4 that it worked acceptably for the images to be used.  Thus, this paragraph should just leave the reader with the impression that a standard method has been used and that the figure confirms that it works.  I like the wording about “laying a solid foundation for image fusion.”

L330-331 “ensure the integrity of the selected features” doesn’t seem to me like an ideal description of the reason for rotating the image.  Surely the real reason is that the curvelet transformation operates on a rectangular domain (of lines and samples), so the strips must be rotated to make the overlap area such a rectangle.

L 333-334 “areas 1-3 were selected” is potentially confusing.  This is the passage that caused me to wonder if the sharpness assessment was performed locally for different areas such as these.  I think what is meant, though, is just that the whole overlap was processed, and then these areas were selected to be enlarged in the figures because they illustrate the results well.

Equation 17 I think the values in the resolution vectors might be presented more effectively in a table than as a pair of equations.

Equation 18 is not needed – it is the same as Equation 14 above, so the authors could simply say “the ratio, computed according to Eq. 14, is 0.1428.”

L 378ff  The three “evaluation indices” sound rather similar to some of the sharpness measures.  Are they identical or modified?  Can the authors comment on why these three indices were chosen for evaluation, and a larger set was chosen for the resolution vector?  By the way, this line of thought points to an obvious direction for future research:  would be better results be obtained with a different set of resolution measures?  This could include adding other measures, eliminating ones that do not help the result, or even modifying some of the equations.

Figure 6  This is a well-crafted figure showing the comparison, but I would have a very hard time deciding which of these results I would consider “best.”  I can see that they differ in subtle ways, but they all seem to have similar resolution.  Possibly the “traditional curvelet fusion” result has a little less noise than the others without seeming to sacrifice resolution.

L 425-427 I suggest deleting a phrase so that “the proposed method can effectively eliminate the displacement and distortion of images and greatly improve the fusion results” becomes “the proposed method can greatly improve the fusion results”.  As already discussed, the authors are not proposing or testing in detail a new method for eliminating displacements.  They are only testing the ability to improve on the fusion of images once they are registered by an existing, straightforward method.

The Conclusion section is quite short and could be expanded if the authors desired.  This would be a good place to discuss some of the issues that I have raised in this review, such as how the method is likely to perform on more/less favorable data, whether it might be useful for images other than sonograms, and ways in which the fusion approach might be further improved and tested in the future.

Reviewer 3 Report

Comments to the Author

The paper proposed an image mosaic method based on Curvelet transform and resolution constraints. Experimental results show the performance of the proposed method is superior to the existing methods. However, some issues should be addressed before the publication. Especially, the experimental part needs to be strengthened.

Major issues:

1) The structure of the paper is not standardized. The introduction part does not list the contributions of this paper. What is the significance of the description in Section 2? Is it related work? I think the Section 2 and Section 3.1 should belong to the related work. The proposed method is in Section 3.2. However, the proposed method is only introduced through the schematic diagram in Figure 2, which is too simple. The authors should reorganize the structure of the paper and introduce this part in detail.

2) There is no corresponding literature for the comparison methods, and the comparison methods is also not novel. It is suggested that the authors further add the relevant state-of-the-art comparison algorithms.  

3) For all the methods, the experimental operation time index should be given.

4) Only one set of experimental data is used in this paper, which can not generally prove the advantages of the proposed method. It is recommended to add at least one more set of experimental data.

Minor issues:

1) The methods proposed by the authors include many image processing methods based on machine learning, such as registration, super-resolution, fusion and so on. So, in the introduction part, a comprehensive and systematic background introduction and description related to the advanced and latest machine learning works, e.g.,

[1] Registration of Multiresolution Remote Sensing Images Based on L2-Siamese Model, IEEE Journal of Selected Topics in Applied Earth Observations and Remote Sensing, 2021, 14: 237 - 248.

[2] Super-Resolution Mapping Based on Spatial-Spectral Correlation for Spectral Imagery [J]. IEEE Transactions on Geoscience and Remote Sensing, 2021, 59(3): 2256-2268.

2) The last part of the introduction needs to give chapter arrangement.

3) The English could be further improved. There are many grammatical errors in the article. In addition, Figure 1 is not clear enough. It looks like a network screenshot rather than the author himself. It is suggested that the author redraw it himself.

Reviewer 4 Report

Globally, the manuscript is very well written and organized.

English needs small corrections; please refer to the attached document, where some of the needed corrections are highlighted in yellow.

I suggest the authors to include a sentence, in section 2, stating that the image resolution assessment methods presented in this section will all be used by the proposed method. I also recommend to present the main reasons for using these assessment methods and no other.

At the beginning of sub-subsection “2.2.2. Discrete Cosine Transform (DCT)” you state “DFT-based resolution assessment has high sensitivity, however, the DFT coefficient is a complex, resulting in a large calculation amount.”. Do you want to say that the DFT is computationally more demanding than DCT? (because it is). If so, please rephrase this sentence and say it clearly.

Corrections in the numbering of sections, starting from section 3, should be introduced. Please refer to the attached document, where this is clearly stated.

I also recommend the inclusion of future working directions and improvements in the proposed method, in the conclusions section.

Some other minor corrections are also noted in the attached document.

Round 2

Reviewer 3 Report

Thank you for your reply. I have no more questions.

This manuscript is a resubmission of an earlier submission. The following is a list of the peer review reports and author responses from that submission.